

# FDTD analysis of ELF radio wave propagation in the spherical Earth-ionosphere waveguide and its validation based on analytical solutions

Volodymyr Marchenko[1], Andrzej Kulak[2], Janusz Mlynarczyk[2]

[1]Jagiellonian University, Astronomical Observatory, Krakow 30-244, Poland

[2]AGH University of Science and Technology, Institute of Electronics, Krakow 30-059, Poland

*Correspondence to*: Volodymyr Marchenko (volodymyr.marchenko@oa.uj.edu.pl)

**Abstract.** The FDTD model of electromagnetic wave propagation in the Earth-ionosphere cavity was developed under assumption of axisymmetric system, solving the reduced Maxwell's equations in a 2D spherical coordinate system. The model was validated on different conductivity profiles for the electric and magnetic field components for various locations on Earth

along the meridian. The characteristic electric and magnetic altitudes, the phase velocity and attenuation rate were calculated. We compared the results of numerical and analytical calculations and found good agreement between them. The undertaken FDTD modeling enables us to analyze the Schumann resonances and the propagation of individual lightning discharges occurring at various distances from the receiver. The developed model is particularly useful when analyzing ELF measurements.

**Keywords.** ELF radio wave propagation, FDTD method, Earth-ionosphere waveguide, Schumann resonances, decomposition method.

## 1 Introduction

The finite-difference time-domain (FDTD) is a numerical analysis technique based on time-dependent differential Maxwell's equations. It was originally developed for Cartesian coordinate system, but after elaboration of the code for spherical

coordinates, it found applications in studies of ELF and VLF radio wave propagation in the Earth-ionosphere waveguide (Holland, 1983; Hayakawa and Otsuyama, 2002; Simpson and Taflove, 2002; Otsuyama et al., 2003; Yang and Pasko, 2005; Yu et al., 2012; Samimi et al., 2015). Similar analysis can be performed for Mars and other planets (Soriano et al., 2007; Navarro et al., 2007; Yang, et al., 2006; Navarro et al., 2008).

When a small part of the Earth-ionosphere cavity needs to be analyzed, a local volume can be divided into FDTD grid in

2D cylindrical (Cummer, 2000; Hu and Cummer, 2006; Qin et al., 2019), or 3D Cartesian coordinate systems (Araki et al., 2018; Suzuki et al., 2016). It facilitates taking into account some complex inhomogeneities and ionospheric anisotropy in the analysis of ELF/VLF radio waves.



FDTD modeling in 3D Cartesian coordinates system was used for verification of Wait's and Cooray-Rubinstein analytical formulas, describing lightning-radiated electric and magnetic fields for a mixed propagation path (vertically stratified conductivity) and for the fractal rough ground surface (Zhang et al., 2012a; Zhang et al., 2012b; Li et al., 2013; Li et al., 2014).


The analysis of propagation of over a mountainous terrain was performed in 2D axial symmetric model (Li et al., 2016), which was further developed into FDTD model in 2D spherical axisymmetric coordinate system (Li et al., 2019). In such modeling the authors have investigated the effect of the Earth-ionosphere waveguide structure and medium parameters, including the effect of the ionospheric cold plasma characteristics, the effect of the Earth curvature, and the propagation effects over a mountainous terrain.


The influence of ionospheric disturbances on the Schumann resonances was analyzed using 3D FDTD model in (Navarro et al., 2008). The authors estimated the role of day-night asymmetry, polar non-uniformities associated to solar proton events, and X-ray bursts.

Few other numerical approaches were used for the estimation the parameters of Earth-ionosphere resonator, like the two-


dimensional telegraph equation (TDTE) in a two-dimensional spherical transmission line model of the Earth-ionosphere cavity (Kulak et al., 2003), Transmission Line Matrix (TLM) numerical method in Cartesian and spherical coordinate systems (Morente et al., 2003; Toledo-Redondo et al., 2016).

Besides the Schumann resonances, another important aspect of ELF studies concerns propagation of ELF waves generated by individual lightning discharges. Their waveforms observed at large distance from the source are significantly influenced by


the dispersive properties of the Earth-ionosphere waveguide and the over-the-world propagation. Inverse solutions in such case should take into account full non-uniform solutions of the Maxwell equations. This is particularly important for lighting discharges that have a long continuing current phase or are associated with Transient Luminous Events (TLE) when they occur at large distance from the receiver.

The development of an FDTD model was motivated by our ELF systems: World ELF Radiolocation Array (WERA) (Kulak


et al., 2014) and a new European ELF radiolocation system (EERS) (Mlynarczyk et al., 2018), which are operating in the Extremely Low Frequency (ELF) range, as well as for possible stations on Mars.

In this paper, we present an FDTD uniform model in 2D axisymmetric spherical coordinates and introduce new approach for validation the FDTD models by introducing computation of complex altitudes of the Earth-ionosphere waveguide, which allows us to compare numerical results with two-scale-height analytical solutions. We also infer the resonance frequencies of


Earth-ionosphere waveguide for various conductivity profiles using the decomposition method (Kulak et al., 2006).

## 2 FDTD model

We have created the FDTD model following the ideas, which were originally proposed in (Holland, 1983) and developed in further studies (Hayakawa and Otsuyama, 2002; Otsuyama et al., 2003; Yang and Pasko, 2005; Navarro et al., 2007; Yang, et



al., 2006; Navarro et al., 2008). We chose a spherical coordinate system to be able to study the wave's propagation in the
Earth-ionosphere cavity and analyze the Schumann resonances.

Since in the present work we did not intended to study the azimuthal dependence of propagation parameters, we reduced a
3D system of Maxwell's equations to a 2D axisymmetric system. It allowed us to significantly decrease the required
computational power and enabled longer simulation times, which leads to a better frequency resolution (up to df = 0.001 Hz).

### 2.1 Update equations

In the case of axisymmetric system, assuming no dependence on $\varphi$ coordinate, the system of six Maxwell's equations in 3D
spherical coordinates ($r$, $\theta$, $\varphi$) can be reduced to 2D spherical system ($r$, $\theta$) with tree equations for $E_r$, $E_\theta$, and $H_\varphi$ field
components (Holland, 1983; Inan and Marshall, 2011)

$$\varepsilon_0 \frac{\partial E_r}{\partial t} + \sigma E_r + J_r = \frac{1}{r\sin\theta}\left[\frac{\partial}{\partial\theta}\left(H_\varphi \sin\theta\right)\right], \tag{1}$$

$$\varepsilon_0 \frac{\partial E_\theta}{\partial t} + \sigma E_\theta = -\frac{1}{r}\frac{\partial}{\partial r}\left(rH_\varphi\right), \tag{2}$$

$$\mu_0 \frac{\partial H_\varphi}{\partial t} = -\frac{1}{r}\left[\frac{\partial}{\partial r}\left(rE_\theta\right) - \frac{\partial E_r}{\partial\theta}\right], \tag{3}$$

where $\sigma = \sigma(r)$ is the conductivity profile of Earth-ionosphere cavity, $\varepsilon_0$ and $\mu_0$ are the vacuum permittivity and
permeability respectively.

These equations were discretized using central-difference approximations to the space and time partial derivatives. The
resulting finite-difference equations are solved in a leapfrog manner: the electric field vector components in a volume of space
are solved at a given instant in time; then the magnetic field vector components in the same spatial volume are solved at the
next instant in time and the process is repeated over and over again until the desired transient or steady-state electromagnetic
field behaviour is fully evolved (Inan and Marshall, 2011). So the update equations for $E_r$, $E_\theta$ and $H_\varphi$ are the following
(Holland, 1983)





$$E_r\,|_{i,j}^{n+1/2} = \frac{\left(\varepsilon_0/\Delta t - \sigma_{i,j}/2\right)}{\left(\varepsilon_0/\Delta t + \sigma_{i,j}/2\right)} E_r\,|_{i,j}^{n-1/2} +$$

$$\frac{1}{\left(\varepsilon_0/\Delta t + \sigma_{i,j}/2\right)} \left( -J_r\,|_{i,j}^{n} + \frac{1}{r_{i+1/2}\sin\theta_j} \right.$$

$$\left. \left[ \frac{\sin\theta_{j+1/2}H_\varphi\,|_{i,j}^{n} - \sin\theta_{j-1/2}H_\varphi\,|_{i,j-1}^{n}}{\Delta\theta} \right] \right), \qquad (4)$$

$$E_\theta\,|_{i,j}^{n+1/2} = \frac{\left(\varepsilon_0/\Delta t - \sigma_{i,j}/2\right)}{\left(\varepsilon_0/\Delta t + \sigma_{i,j}/2\right)} E_\theta\,|_{i,j}^{n-1/2} +$$

$$\frac{1}{\left(\varepsilon_0/\Delta t + \sigma_{i,j}/2\right)r_i} \left[ \frac{r_{i+1/2}H_\varphi\,|_{i,j}^{n} - r_{i-1/2}H_\varphi\,|_{i-1,j}^{n}}{\Delta r} \right], \qquad (5)$$


$$H_\varphi\,|_{i,j}^{\,n+1} = H_\varphi\,|_{i,j}^{n} - \frac{\Delta t}{\mu_0 r_{i+1/2}}$$

$$\left[ \frac{r_{i+1}E_\theta\,|_{i+1,j}^{n+1/2} - r_i E_\theta\,|_{i,j}^{n+1/2}}{\Delta r} - \frac{E_r\,|_{i,j+1}^{n+1/2} - E_r\,|_{i,j}^{n+1/2}}{\Delta\theta} \right], \qquad (6)$$

where $\Delta t$ is the time step, $\Delta r$ and $\Delta\theta$ are the sizes of grid cell in $r$ and $\theta$ coordinates respectively, $r_i = R_0 + i\Delta r$, $r_{i\pm1/2} = R + (i\pm1/2)\Delta r$, $\theta_j = j\Delta\theta$, $\theta_{j\pm1/2} = (j\pm1/2)\Delta\theta$, $R_0$ is the mean Earth's radius. Superscript $n$ signifies that the

quantities are to be evaluated at time $t = n\Delta t$, and $i$, $j$ represent the point ($r = i\Delta r$, $\theta = j\Delta\theta$) in the spherical grid. The half time steps indicate that the electric and magnetic fields are calculated alternately.

The update equation for $E_r$ cannot be applied for poles (where $\theta = 0$ and $\theta = \pi$) because of $\sin\theta$ in denominator. To solve this singularity problem the Holland's approach was used in Holland (1983). Namely, for the poles the integral form of the curl equation with a small contour around the poles was applied, which leads to new update equations for poles without

singularities. After that, the update equation for $\theta = 0°$ takes the form (Holland, 1983):

$$E_r\,|_{i,j}^{n+1/2} = \frac{\left(\varepsilon_0/\Delta t - \sigma_{i,j}/2\right)}{\left(\varepsilon_0/\Delta t + \sigma_{i,j}/2\right)} E_r\,|_{i,j}^{\,n-1/2} +$$

$$\frac{1}{\varepsilon_0/\Delta t + \sigma_{i,j}/2} \left[ -J_r\,|_{i,j}^{n} + \frac{\sin\theta_{1/2}H_\varphi\,|_{i,0}^{n}}{r_{i+1/2}(1-\cos\theta_{1/2})} \right]. \qquad (7)$$





And for $\theta = 180°$ :


$$E_r \mid_{i,j}^{n+1/2} = \frac{\left(\varepsilon_0/\Delta t - \sigma_{i,j}/2\right)}{\left(\varepsilon_0/\Delta t + \sigma_{i,j}/2\right)} E_r \mid_{i,j}^{n-0.5} +$$

$$\frac{1}{\varepsilon_0/\Delta t + \sigma_{i,j}/2}\left[-J_r \mid_{i,j}^{n} - \frac{\sin\theta_{N_\theta-1/2}H_\varphi \mid_{i,N_\theta-1}^{n}}{r_{i+1/2}(1+\cos\theta_{N_\theta-1/2})}\right],$$

(8)

where $\theta_{1/2} = \Delta\theta/2$, $\theta_{N_\theta-1/2} = (N_\theta - 1/2)\Delta\theta$, $N_\theta$ is the number of grid cells in $\theta$ direction.

### 2.2 Mesh

The size of grid cell (in $r$ and $\theta$ directions) is defined by the desired range of frequencies, which are going to be analyzed.

For the present analysis we used $\Delta r = 1$ km and $\Delta\theta = 0.1°$, which lets us analyze frequencies up to 1 kHz. The ground was

modeled as a perfect electric conductor. Also, the upper boundary $R_{max}$ for the model is a perfect conductor but it was placed

at high enough altitude to make sure there is no reflection of the waves from this boundary.

The total simulation time is defined by the desired frequency resolution of FFT ($\Delta f = 1/t_{max}$ Hz). The time step is defined

by the stability Courant's criterion $\Delta t \leq 1/c\sqrt{(\Delta r)^{-2} + (a\Delta\theta)^{-2}}$ , where $c$ is the speed of light and $a$ is the Earth radius (Inan

and Marshall, 2011).

In order to get sufficient frequency resolution (in our case it's 0.001 Hz, to be able to study the differences between different

models) we would have to run the FDTD simulation up to 1000 s. Due to reduction of Maxwell's equations from 3D to 2D we

were able to run simulations up to 100 s. For further improvement of frequency resolution we have extended the simulation

time to 1000 s by adding the values of fields at the last time step of FDTD simulation. Since the field amplitudes are negligibly

small after 100 s of simulation, such extension does not produce significant distortions of FFT result.

### 2.3 Source

In order to analyze the propagation of electromagnetic waves originating from lightning discharges, one has to use the source

model with similar characteristic to real lightning discharge. We used the time profile of lightning discharge proposed in

(Kulak et al., 2010; Rakov, 2007), which has a flat spectrum in a wide frequency range. But taking into account the restrictions

connected with FDTD computational grid, we had to modify the source – the highest frequency (or smallest wavelength) is

defined by the cell size, and according to the generally accepted rule the smallest wavelength should be at least 10-20 times





larger than the cell size. Therefore, the source has to be filtered with a loss-pass filter in order to remove frequency components that does not fit the size of computational cell.

Also, it should be noted, that if the source contains the direct current (DC) in its spectrum, it can introduce artifacts, which are not physical (Li et al., 2013). Therefore, additional filtering has to be used for the lowest frequencies by a high-pass filter. The final spectrum of this modified source is almost flat in the range 5–100 Hz but decreases rapidly for $f < 3$ Hz and $f > 1000$ Hz.

We placed the source in the FDTD grid at the pole ($\theta = 0°$) in radial direction in nodes with $i = 0 - 6$, which corresponds

to the source length $L = 6$ km. Also, for the implementation of the source into the FDTD grid we took into account that the lightning stroke has a finite speed (Rakov, 2007). We assumed a cloud-to-ground (CG) discharge and implemented it by the time delay between adjacent nodes in the source.

### 3. Validation of the model

The main purpose of the current work is the analysis of resonance phenomenon in the Earth-ionosphere cavity. In order to

validate the developed FDTD model, we analyzed several configurations of spherical waveguide with known analytical solutions and compared it with our FDTD results. We compared the resonance frequencies for them in relative and absolute units.

### 3.1 Lossless spherical waveguide

The first model for testing was a lossless spherical waveguide with perfect electric conductors at the ground and the upper boundary. The distance between the conductors was set to $h = 74$ km. The precise theoretical resonance frequencies for such waveguide for a given height were presented by the equation $f_n = (c/2\pi a)\sqrt{n(n+1)}\sqrt{1 - h/a}$ (Bliokh et al., 1977). A comparison of analytical and numerical results for such model is presented in Table 1.


Table 1

Comparison of resonance frequencies obtained analytically and numerically for a lossless spherical waveguide.

| Mode | Analytical [Hz] | FDTD [Hz] | Abs.Err [Hz] | Rel.Err [%] |
|------|-----------------|-----------|--------------|-------------|
| $f_1$ | 10.530 | 10.525 | 0.005 | 0.04 |
| $f_2$ | 18.238 | 18.250 | 0.012 | 0.07 |
| $f_3$ | 25.792 | 25.795 | 0.003 | 0.01 |



### 3.2 Spherical waveguide with a conducting layer

A more complicated model for testing had a perfect conductor at the ground and a constant conductivity $4 \cdot 10^{-6}$ S/m above 70 km. The analytical solutions for this model were obtained by Kulak and Mlynarczyk (2013). A comparison of analytical and numerical results for such model is presented in Table 2.

Table 2

Comparison of resonance frequencies obtained analytically and numerically

for a spherical waveguide with a conducting layer above 70 km

| Mode | Analytical [Hz] | FDTD [Hz] | Abs.Err [Hz] | Rel.Err [%] |
|---|---|---|---|---|
| $f_1$ | 8.01 | 8.00 | 0.01 | 0.13 |
| $f_2$ | 14.91 | 14.83 | 0.08 | 0.56 |
| $f_3$ | 23.69 | 21.71 | 0.02 | 0.09 |

### 3.3 Two-layered spherical waveguide

The third model for testing was a two-layered model with a perfect conductor at the ground, a constant conductivity $5 \cdot 10^{-7}$ S/m at altitudes between 70 and 110 km, and a constant conductivity $5 \cdot 10^{-5}$ S/m above 110 km. The analytical results for such model can be obtained from Kulak et al. (2013), which deal with multi-layer waveguides. A comparison of analytical and numerical results for this model is presented in Table 3.

Table 3

Comparison of resonance frequencies obtained analytically and numerically for a two-layered spherical waveguide.

| Mode | Analytical [Hz] | FDTD [Hz] | Abs.Err [Hz] | Rel.Err [%] |
|---|---|---|---|---|
| $f_1$ | 8.007 | 8.003 | 0.004 | 0.05 |
| $f_2$ | 14.084 | 14.09 | 0.006 | 0.04 |
| $f_3$ | 20.078 | 20.07 | 0.008 | 0.04 |





## 4. Application of a realistic atmospheric conductivity profile

The next validation of our FDTD model was done for a realistic atmospheric conductivity profile (Kudintseva et al., 2016; Nickolaenko et al., 2016). We did not take into account the influence of Earth magnetic field. The resulting anisotropy of the conductivity has little influence in the analyzed frequency range (i.e., 0-100 Hz). As it was shown in Yu et al. (2012), the anisotropy has a more significant influence at higher frequencies.

For the lower part of atmosphere (0-100 km) we used a conductivity profile recently proposed in Kudintseva et al. (2016).

Since the upper boundary of this profile has the conductivity much below 1 S/m, which is not enough to attenuate the ELF waves, reflections in FDTD grid would occur from the PEC at its upper boundary. To avoid reflections and let the waves attenuate gradually at high altitudes we extended this profile up to 200 km using the IRI model. We chose it in such a way that the combined profile is smooth (Figure 1). The required IRI profile was found on January 22, 2006, at 6:00 UT for the location typical for mid-latitudes (49N, 23E).


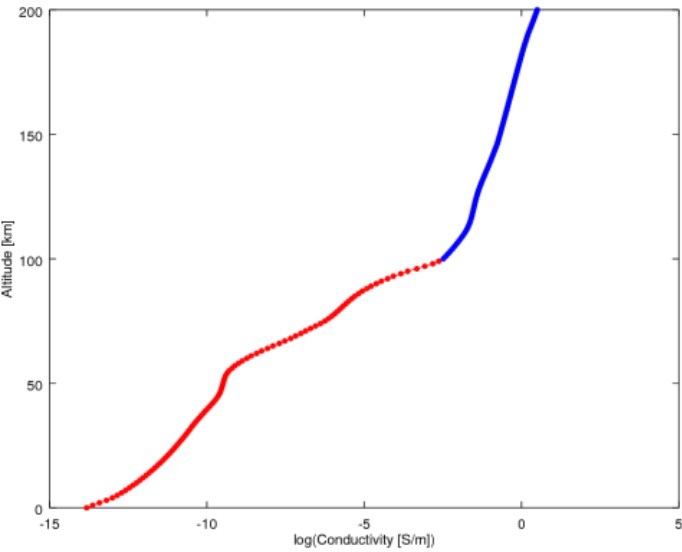

**Figure 1.** A realistic conductivity profile up to 200 km obtained by combining the profile from Kudintseva et al. (2016) (red part) and an IRI profile (blue part).

**4.1 Characteristic electric and magnetic altitudes**

To be able to compare the numerical results with the analytical solutions, we have extracted the complex characteristic electric and magnetic altitudes from the FDTD model. The corresponding altitudes can be extracted using their definition (Mushtak and Williams, 2002; Kirillov, 1993):



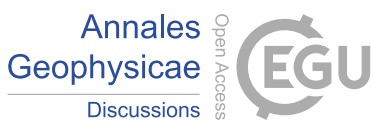

$$\overline{h}_e(f) = \frac{1}{\overline{E}_r^0} \sum_{i=0}^{N} \overline{E}_r^i \Delta r,$$ (9)

$$\overline{h}_m(f) = \frac{1}{\overline{H}_\varphi^0} \sum_{i=0}^{N} \overline{H}_\varphi^i \Delta r,$$ (10)

where $N$ is number of radial nodes, $\overline{E}_r^i$ and $\overline{H}_\varphi^i$ are the complex field values at the radial node $i$ for a given frequency, and

$\overline{E}_r^0$, $\overline{H}_\varphi^0$ are the values of these fields at the ground level.

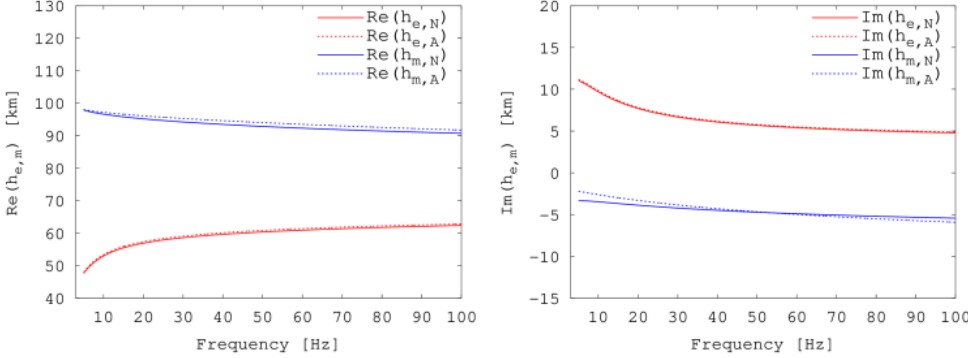

**Figure 2.** Real and imaginary parts of characteristic electric and magnetic altitudes obtained analytically (denoted by "A") and from the FDTD model
(denoted by "N").

The complex electric altitude can also be expressed using the conductivity profile by the analytical equation in a normalized form (Mushtak and Williams, 2002)

$$\overline{h}_e = \int_0^\infty \frac{dr}{1 - i\sigma_e},$$ (11)

where $\sigma_e = \sigma/\sigma_e^*$ and the characteristic conductivity value for electric altitude is $\sigma_e^* = \omega\varepsilon_0$, $\omega = 2\pi f$.

Assuming a similar dependence for complex magnetic altitude and taking into account that the characteristic conductivity value for magnetic altitude is (Greifinger and Greifinger, 1978)






$$\sigma_m^* = \frac{1}{4\mu_0\omega\zeta^2(r)},$$

where the scale height $\zeta(r) = \sigma(r)/(d\sigma/dr)$, we can write a similar equation for the complex magnetic altitude

$$\bar{h}_m = \int_0^\infty \frac{dr}{1+i\sigma_m},$$ (12)

where $\sigma_m = \sigma/\sigma_m^*$. We compared the characteristic electric and magnetic altitudes calculated analytically from equations (11) and (12) with the FDTD results, calculated from equations (9) and (10). The results are presented in Figure 2.

**4.2 Propagation parameters**

We calculated the phase velocity $V_{ph}$ and the attenuation rate $\alpha$ using two different methods – numerical and analytical. Analytically the propagation parameters can be calculated using conductivity profile and electric and magnetic altitudes given by equations (11) and (12), through the following relationship (Kulak and Mlynarczyk, 2011)

$$V_{ph} = \frac{c}{\mathrm{Re}\,\bar{S}},$$ (13)

$$\alpha = \frac{\omega}{c}\mathrm{Im}\bar{S},$$ (14)

where $\bar{S}^2 = \bar{h}_m/\bar{h}_e$.

In case of numerical calculations those parameters were obtained by two approaches: a) similarly to analytical as described above but using electric and magnetic altitudes from FDTD model by equations (9), (10), and b) directly from the spectra of electric and magnetic field components. Assuming that in our coordinate system the wave propagates in the $\theta$ direction the following relationship can be written in the frequency domain

$$\frac{\bar{H}_{\varphi 1}}{\bar{H}_{\varphi 2}}\sqrt{\frac{\sin(\rho_1/a)}{\sin(\rho_2/a)}} = \exp(\bar{\gamma}(\rho_2 - \rho_1)),$$ (15)





where the ratio of magnetic field complex amplitudes $\overline{H}_\varphi$ are calculated for two probes "1" and "2" which are located at distances $\rho_1$ and $\rho_2$ from the source respectively. In this equation the propagation constant $\overline{\gamma} = \alpha + i\beta$ , where $\alpha$ is the attenuation rate (in units Np/m) and $\beta$ is the phase constant, so the phase velocity is $V_{ph} = \omega/\beta$ . For further convenient usage

we convert the units of attenuation rate to dB/Mm. Similar relationship to (15) but for electric field components $\overline{E}_r$ can be written as well.

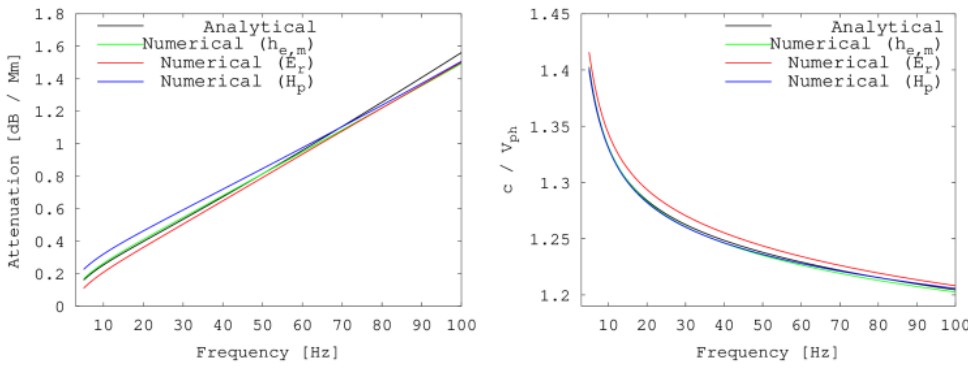

**Figure 3.** The attenuation rate and phase velocity that was calculated analytically and numerically. Using numerical FDTD calculations those parameters were obtained by two approaches: a) using electric and magnetic altitudes from FDTD by equations (9), (10), and b) directly from the spectra electric and magnetic following the equation (15), where we used probes located at $80^0$ and $90^0$.

We should note that equation (15) for $\overline{E}_r$ and $\overline{H}_\varphi$ spectra calculated in closed cavity gives non monotonic behavior of

propagation parameters. This is caused by superposition of wave attenuation along meridian and different amplitudes of the Schumann resonances for different source-observer distance. One of possible solutions for removing the influence of Schumann resonances is to implement a PML along $\theta$ direction. However, we solved this problem in a different way, transforming the Maxwell equations at angle $\theta > 90^0$ from spherical coordinates to plane equations, changing the behavior of equations from "close" to "open". In this case the waves are unable to propagate around the Earth and therefore the Schumann

resonance does not occur. The calculated propagation parameters from all the methods listed above are presented in Figure 3.

### 4.3 Spectral decomposition method

As an additional validation we applied the spectral decomposition technique to Schumann resonance power spectra. Figure 4 presents the resonance frequencies obtained from the spectra measured at different distance from the source, and the decomposed frequencies that do not depend on the source-observer distance. Following the decomposition algorithm (Kulak

et al., 2006; Dyrda et al., 2014) the spectrum is approximated by the function





$$W(f) = s + \frac{z}{f^m} + \sum_{k=1}^{3} \frac{p_k\left[1 + e_k\left(f - f_k\right)\right]}{\left(f - f_k\right)^2 + \left(g_k/2\right)^2},$$ (16)

where $W(f)$ is the signal power spectrum, $s$ is the white noise component, $z/f^m$ is the color noise term, $p_k$ is the power

parameter of the $k$-th resonance peak, $e_k$ is the peak asymmetry parameter, $f_k$ is the resonance frequency, and $g_k$ is the

resonant mode half-width parameter. Fitting this function to the FDTD spectra allows us to extract the resonance frequencies

$f_k$ of the cavity, which are not equal to Schumann resonance frequencies. The Schumann resonance frequencies obtained from

the spectra depend on the distance from the source and represent the superposition of standing and traveling waves (Kulak et

al., 2006). To analyze the standing waves separately and reveal the properties of the resonant cavity we are using the spectral

decomposition method described in (Kulak et al., 2006; Dyrda et al., 2015). After applying this method, the resonance

frequencies become independent of the source-observer distance (see details in Kulak et al. (2006) and Dyrda et al. (2015)).

The decomposition method shows that the solutions for the electric field are symmetric at $\theta = 180°$ because the traveling

waves cancel each other and only the standing waves remain. Therefore, the resonant peaks measured from the spectrum at

$\theta = 180°$ for the electric field component ( $E_r^{180}$ ) represent the resonance frequency of the cavity and they are in agreement

with the decomposed frequencies.

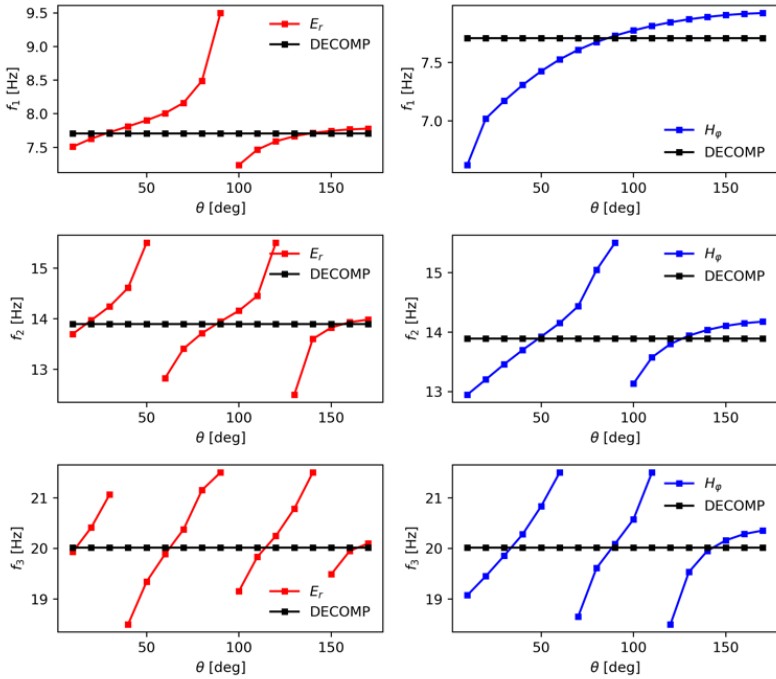

**Figure 4.** The Schumann resonance frequencies of first three modes obtained at different source-observer distances and the resonance frequencies of the cavity obtained by the decomposition method (Kulak et al., 2006).





### 4.4 Resonance frequencies of the Earth-ionosphere cavity

The resonance frequencies of the Earth-ionosphere cavity for a given conductivity profile can be calculated by different approaches and the consistency between them can be considered as an additional validation of the model. We have considered the followings ways to calculate the resonant frequencies:

1). Analytically using the conductivity profile. The resonance frequencies in this approach are calculated by solving the
following equation (Mushtak and Williams, 2002; Galejs, 1972)

$$f_c^n = \frac{c}{2\pi a} \sqrt{n(n+1)} \frac{\mathrm{Re}\,\bar{S}(f_c^n)}{|\bar{S}(f_c^n)|}, \tag{17}$$

where $\bar{S}^2 = \bar{h}_m/\bar{h}_e$ depending on the characteristic complex altitudes, which are discussed in Section 4.1;

2). Numerically from FDTD model using equation (17). We denote these resonant frequencies by $f_h^n$ ;

3). Using FDTD spectra for $E_r^{180}$ (see Section 4.3 for details). We denote these frequencies by $f_{E_r}^n$ ;

4). Using spectral decomposition method (see Section 4.3 for description). We denote these resonant frequencies by $f_d^n$ .

The obtained resonance frequencies are presented in Table 4. These frequencies we compared with the analytical results
and the absolute and relative differences are presented in the corresponding columns.

Table 4

Resonance frequencies for the conductivity profile shown in Figure 1 calculated by different approaches described in Section 4, and the relative error obtained by comparison with the analytically obtained resonance frequency.


| Mode | $f_c^n$ [Hz] | $f_h^n$ [Hz] | $f_{E_r}^n$ [Hz] | $f_d^n$ [Hz] | $\Delta f_h$ [Hz / %] | $\Delta f_{E_r}$ [Hz/%] | $\Delta f_d$ [Hz %] |
|---|---|---|---|---|---|---|---|
| $f_1$ | 7.711 | 7.705 | 7.783 | 7.707 | 0.00565 / 0.073 | 0.072 / 0.936 | 0.004 / 0.052 |
| $f_2$ | 13.877 | 13.909 | 13.998 | 13.892 | 0.0324 / 0.234 | 0.121 / 0.872 | 0.015 / 0.105 |
| $f_3$ | 19.968 | 20.070 | 20.140 | 20.017 | 0.102 / 0.511 | 0.172 / 0.861 | 0.049 / 0.247 |
| $f_4$ | 26.049 | 26.246 | 26.275 | 26.160 | 0.196 / 0.753 | 0.226 / 0.866 | 0.111 / 0.425 |
| $f_5$ | 32.140 | 32.448 | 32.409 | 32.351 | 0.308 / 0.960 | 0.269 / 0.838 | 0.211 / 0.658 |



### 5. Discussion

In this study, we used new methods for validation of numerical simulation:

1). We compared the complex electric and magnetic altitudes of the Earth-ionosphere waveguide, referring to two-

dimensional formalism of electromagnetic wave propagation in the Earth-ionosphere cavity. The two complex altitudes were calculated numerically directly from their definitions (9) and (10), making use of radial field solutions $E(r)$ and $H(r)$. These altitudes were directly compared with the altitudes obtained with analytical formulas (11) and (12) for the same conductivity profile of the atmosphere. This allowed us to fully validate the simulation results.

2). We determined the resonance frequencies of the cavity using the decomposition of power spectra described by equation

(16). The resonance frequencies cannot be directly determined from the spectra obtained using FDTD, because the field generated by the source in each point of the cavity is a superposition of travelling waves propagating directly from the source and the resonance field resulting from the interference of waves propagating around the world. The resulting Schumann resonance frequencies depend on the source-observer distance (Kulak et al., 2006). Close to the source, where the amplitudes of the travelling waves are significant, the resonance frequencies differ by several percent from the Schumann resonance

frequencies obtained by FDTD. With the use of the decomposition method we determined the intrinsic resonance frequencies of the cavity, which are the same at each location, independently from the distance to the source. This enabled us to compare the resonance frequencies obtained from the numerical simulation with the frequencies determined directly from analytical formula (9). This method should be recommended as a reference for validation of numerical models.

3). We determined the propagation parameters of the Earth-ionosphere waveguide using the FDTD results in two different

points of the great circle. Equation (15) that we used allows us to determine the phase velocity and the attenuation rate of the Earth-ionosphere cavity. We compared them with the results obtained from analytical formulas (13) and (14).

### 6. Summary and conclusions

In this paper, we analyzed the solutions of Maxwell's equations obtained by the FDTD method for an axisymmetric uniform Earth-ionosphere cavity. We analyzed the propagation of radio waves generated by a short current impulse.

We have constructed an FDTD model in axisymmetric spherical coordinate system with the source implemented at the pole. We took into account the finite speed of lightning discharge and implemented the time delay between adjacent nodes in the source.

We validated the model thoroughly, comparing the resonance frequencies, propagation parameters and electric and magnetic characteristic altitudes. Since the conductivity profile of the atmosphere has a significant influence on radio wave

propagation and resonance frequencies, we validated our model for various conductivity profiles.

We paid a close attention to the verification of accuracy of the FDTD computations and used a new approach for that purpose. It is based on 2D formalism of wave propagation in the Earth-ionosphere cavity and it allowed us to compare the numerical and the analytical solutions. In this approach, the propagation parameters of the Earth-ionosphere waveguide are defined using the electric and magnetic altitudes. These altitudes can be calculated directly, when the vertical conductivity profile of the atmosphere is known.

The obtained analytical solutions were used as reference and compared with the numerical results. First, we compared the solutions for three cases: a perfect cavity, a cavity formed by a perfect ground and a homogenous conducing layer, and a cavity formed by a perfect ground and two-layered upper boundary of the waveguide. As a measure of error between the models, we took the difference between the first three resonance frequencies. The analytical and numerical solutions were in agreement (Tables 1, 2, 3). Next, we compared the results for a continuous conductivity profile. We built a realistic conductivity model, which lower part was based on a recently proposed conductivity profile and the upper part was based on an IRI model. We obtained a good agreement between the resonance frequencies of the cavity and the observed Schumann resonance frequencies (Table 4).

Using our FDTD model, we also calculated the spectral dependence of the phase velocity and the attenuation rate. We showed that the analytical and numerical models are in agreement.

The presented model can be used for studying the propagation of ELF electromagnetic waves generated by lighting discharges of various types with the round-the-world propagation taken into account.

### Acknowledgments

The numerical computations were done using the PL-Grid infrastructure. The present work was supported by the Polish National Science Center under the grants: 2016/22/E/ST9/00061 (VM), 2015/19/B/ST9/01710 (AK), 2015/19/B/ST10/01055 (JM).

### Author contributions

AK formulated the concept of the current research. VM wrote the FDTD code, the python scripts for the post-processing and visualization, contributed to the development of the analytical model used for the validation of FDTD results in Section 4 and prepared the initial version of the manuscript. JM calculated the analytical results for the validation in Section 3 and participated in the preparation of the scripts for the post-processing of FDTD results. All authors participated in the interpretation of the results and contributed to the preparation of the final version of the manuscript.





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
