# Peer review of "FDTD analysis of ELF radio wave propagation in the spherical Earthionosphere waveguide and its validation based on analytical solutions"

_Annales Geophysicae, 2021_

## Author Comment (AC1)

**Answers to reviews of the paper**
**"FDTD Analysis of ELF Radio Wave Propagation in the Spherical Earth-ionosphere Waveguide and its Validation Based on Analytical Solutions",**
**by V. Marchenko, A. Kulak, and J. Mlynarczyk**

**Reviewer 1**

**General comments**

**REVIEWER COMMENT:**

Judging from the Title, the manuscript must be close to the paper [1] where the 2DTE and the FDTD solutions were compared. However, this is not so. The manuscript title is misleading since the text contains minor TD information and predominantly the approximate expressions in the FD.

**ANSWER:**
Thank you for taking your time to review our manuscript. We did not show the solution in TD, because we were mostly interested in the analysis of our results in FD, which we got by applying FFT to the TD results. We agree that it could be interesting for the reader to see the TD results, therefore we have added a figure showing the results in TD at two different locations along the meridian.We have also added the FD results in the same locations. We can add more TD results if the reviewer finds it useful.

**REVIEWER COMMENT:**

It seems that authors do not not know of the monographs on Schumann resonance (SR), the chapters in handbooks, or recent publications in the field [2–7]. Content of these works is relevant to the manuscript objective, therefore, comparison and discussion are necessary of similarities and departures from the published model results.

**ANSWER:**

Thank you for pointing out these papers. We do know these publications. They are relevant to our field of research, but study [7] is relevant also to this particular manuscript so we added a reference to it in the introduction.

**REVIEWER COMMENT:**

The major problem with the manuscript is the declared time domain solution while the dominant portion of material is presented in the frequency domain. It is unclear what the time domain results look like (these must be similar to waveforms shown in paper [7] or to the pulses presented in monographs [2, 3]). A reader is not informed of how the the TD data were transformed into the specific parameters in the FD. This specific relevance is really interesting when the solution is constructed directly in the time domain.

**ANSWER:**

Thank you for this comment. As we already mentioned in the previous answer, we first solve the equations in TD, calculating the propagation of EM waves in the Earth-ionosphere cavity. Then the obtain the solution in FD domain by applying FFT to the TD results. For illustration we have added a figure showing an example of output FDTD results in TD at two different locations along the meridian.

**Particular remarks and comments**

**REVIEWER COMMENT:**

> The particular drawbacks that should be addressed in revision are obviously seen in the Discussion of the manuscript. I list the points in the correct way.
> 1) The complex characteristic heights being the functions of the signal frequency are discussed in the manuscript. These are used when describing the ELF radio wave propagation in a uniform 3D Earth – ionosphere cavity. Within such cavity, the field components at a particular frequency are the functions of only single variable – the distance between the source and the observer. The problem is a 3D one, and its solution is the function of a single variable. The term "axially symmetric cavity" is irrelevant and actually misleading in the case. Formulas (9) and (10) are similar to the exact relations obtained in the full wave solution; however, the method remains unclear of their obtaining in the time domain.

**ANSWER:**

The problem is indeed 3D, but since in this study we consider a univorm cavity, we can simplify the calculations assuming an axial symmetry. We called this configuration axisymmetric. We have also tested a full 3-D spherical system, with the dependence on two angles $\varphi$ and $\theta$ i we obtained identical results, but the simulation time was of course a lot longer.

**REVIEWER COMMENT:**

> Equations (11) and (12) actually repeat the results formulated by Greifingers (1978) and mentioned by Madden and Thompson (1965) [8]. However, Greifingers were much more cautious than the authors of this manuscript: the heights were obtained numerically using the full wave solution while the formulae were suggested for the physical interpretation of numerical results. Greifingers used the term 'approximate' for these equations.

**ANSWER:**

We would like to emphasize, that equations (11) and (12) in our paper works for arbitrary conductivity profiles, while Greifingers (1978) can only be used for an exponential profile. In this study we obtain the heights both analitycally and numerically and we show that they are in agreement.

**REVIEWER COMMENT:**

> There is only one analytical solution in the theory of Schumann resonance, and this is the Schumann formula for the eigen-frequencies of an ideal cavity. The resonant frequencies of a real Earth–ionosphere cavity are complex, and these are always obtained as a result of calculations. There are no closed rigorous analytical expressions for the complex resonant frequencies. We must note here that the Q-factor of Schumann resonance oscillations ranges from 4 at the first mode to 6 at the fifth mode, see, e.g., Madden and Thompson [8]. When discussing measurements, one turns to the peak frequencies. These are the frequencies of the maxima in the spectral density of ELF radio signal. The peak frequencies are, of course, related to the resonance frequencies, but their value depends also on the source spectrum, the source-observer distance (SOD), and on the local interferences at the observatory. The clear partition of these terms is absent in the text thus leading to confusion and misunderstanding.

**ANSWER:**

We believe that we have clearly separated the concept of the Schumann resonance frequencies and the resonance frequencies. We have been studying this topic for many years and we have already introduced

and discussed it in Kulak et al. 2003 and Kulak et al. 2006. We were first to point out the difference between the Schumann resonance frequencies and true resonance frequencies of Earth-ionosphere cavity. We have developed the decomposition method based on the idea of the Fano resonance, which allows us to obtain the resonance frequencies from the Schumann resonance measurements. In this study we use the decomposition method to validate the solutions of the numerical model. In this manuscript we do not describe all the details of this method, because it was already described in our previous publication (Kulak et al. 2006, Dyrda et al., 2014).

**REVIEWER COMMENT:**

2) All the stuff above is well-known, and it is widely used in literature on sub-ionospheric ELF radio wave propagation. In this context, Eq.(16) of the manuscript is a replica of the explicit formal solution for the ELF radio propagation problem. I reproduce below the zonal harmonic series representation (ZHSR) for the fields from Chapter 14 of [3] ... **[skipped equations]**.

**ANSWER:**
Thank you for this comment, but we can not agree with it. The equation (16) is not by any means a replica of equations (14.12), (14.13). This equation serves a different purpose: it separates the travelling waves from the standing waves. The relevance of equation (16) was shown in section 4.3 of our manuscript and was used before in application to real ELF data (e.g., Dyrda et al., 2014).

**REVIEWER COMMENT:**

3) Equation (15) of the manuscript is a real puzzle. Authors state that they use "...electric and magnetic altitudes from FDTD model by equations (9), (10)..." In what a way the functions of frequency might be used in the time domain model? This looks odd to me, especially, as Eq.(15) excludes the field focusing effect in a spherical cavity, which becomes especially apparent in the TD pulsed amplitude, see the last figure in paper [9]. When the wide-band signals are treated, the TD waveforms might become really complicated, see the recent paper [7] on the tweeks and the slow tails sferics.

**ANSWER:**
These parameters are in frequency domain. To made it more clear in the revised manuscript byt explicitly denoting them as functions of frequency. We have used the solutions obtained in TD using the FDTD method for the calculations of the correspondent solution in FD using FFT, because we were interested in the analysis of dependence of attenuation versus frequency.

**REVIEWER COMMENT:**

Let us leave the Discussion section of manuscript and turn to a few particular places of the text. - Parameters of ELF radio propagation in the uniform isotropic cavity are insensitive to the air conductivity above the 100–110 km altitude. The full wave solution indicated that the contribution from the upper layers is less than 10–7. Authors must indicate for what a purpose the profile was extended in height and what was the effect of this extension.

**ANSWER:**
We have decided to extend the conductivity profile from Kudintseva et al. (2016) with some realistic continuation to higher values of conductivity, because the profile from Kudintseva et al. (2016) reaches at the top the value of conductivity $\sigma << 1$, and we saw that the electromagnetic waves were not attenuated enough, causing artefacts.

**REVIEWER COMMENT:**

> - It was shown in series of papers that the heuristic predetermined frequency dependence of characteristic heights does not correspond to a profile implying the particular heights, the conductivities and the scale heighgts [1, 10–14]. The full wave solution for such profiles provides the complex electric and magnetic heights departing from those postulated in the heuristic formulas. Do the data of Fig. 2 and Fig 3 support these results or not?

**ANSWER:**

If we understand this comment correctly, we would like to highlight, that for our FDTD simulations we have used a realistic conductivity profile as a function of altitude, in order to validate our FDTD calculations and compare it with analytical results. In the Fig.2 we show characteristic electric and magnetic altitudes, which corresponds to analytical models and we found good agreement between numerical and analytical calculations.

**REVIEWER COMMENT:**

> - Spectral resonance characteristics shown in the Tables raise doubts and make a reader suspicious. The resonance frequencies there are shown with an accuracy of about 10–3 Hz. To reach such a resolution, the TD duration must be about 1000 seconds or more than 10 min. Using of such duration is unbelievable in the TD computations.
> In addition, the SR Q-factors are equal to 4 – 6, so that the half-power width of resonance curves is about 2 Hz. How the 10–3 Hz resolution was obtained while the resonance curve is so wide?

**ANSWER:**

We would like to clarify this issue. We were able to run simulations up to 100 s using the computer cluster, which get us frequency resolution of 0.01 Hz.

As additional option, we were trying to improve the frequency resolution by extending the simulation manually to 1000 s by adding the values of EM fields at the last time step of FDTD simulation (these values at $t = 100$ s are almost zero). However, we agree, that such operation is not improving in fact the frequency resolution in case of strong attenuation of electromagnetic waves. So we have decided to remove this manual extension and keep our real resolution that is defined by the simulation time ($t_{max} = 100$ s, $df = 0.01$ Hz). We would like to thank you for pointing this out.

**REVIEWER COMMENT:**

> - The color lines in Fig. 4 show the distance variations of the peak SR frequencies. Such curves were demonstrated for the first time by J. Galejs, and one may found them in his monograph of 1972. These variations are driven by overlapping adjacent modes. Emergence of the discontinuity in Fig. 4a is explained in detail in book [2]. Distance variations of peak frequencies, similar to Fig. 4a and Fig. 4b were used for deducing parameters of the global thunderstorm activity from the SR records. I refer only to the recent work [15] exploiting the long-term SR data from the Antarctic and Arctic observatories.
> It is not a surprise that frequency variations in Fig. 4 look quite realistic. The only question remains: does the specific FDTD data match with the known results obtained for the same conductivity profile?

**ANSWER:**

These plots show the change of Schumann resonance frequencies with distance from the source. And also show the true resonance frequencies, the derivation of which is possible only by the Decomposition method developed by our team.

**REVIEWER COMMENT:**

The last but not the least. Authors casually mention in the text the comparing of their data with the perfect cavity model.
I doubt the prefect cavity was treated in the time domain owing to severe problems arising even in a cavity with the small finite losses, see paper [16].
In case I am wrong and authors succeed in performing TD computations for a perfect cavity, I advise them to plot the pulsed successions at the points of the source and of the source antipode. Such a revolutionary result, if any, is worth of a separate publication.

**ANSWER:**
We are pleased to report that we were able to solve the perfect cavity model.
Below we show the electric field component $E_r$ in the time domain for different locations in the cavity: $\theta = 0° \, 90° \, 180°$.
We also present the spectra for these locations ($90°$ and $180°$).

[Figure]

Figure 1: The time profiles for electric component $E_r$ for different probes $\theta = 0° \, 90° \, 180°$.

[Figure]

Figure 2: The spectrum of electric $E_r$ and magnetic $H_\varphi$ components at position of probe $\theta = 90°$, $180°$
.

We hope that after seeing these results, the reviewer gained full confidence in our FDTD model and our understanding of this research field.

---

## Author Comment (AC2)

**Answers to reviews of the paper**
**"FDTD Analysis of ELF Radio Wave Propagation in the Spherical Earth-ionosphere Waveguide and its Validation Based on Analytical Solutions",**
by V. Marchenko, A. Kulak, and J. Mlynarczyk

**Reviewer 2**

**Major comments:**

**REVIEWER COMMENT:**

> Line 115-116, I don't really understand this part, what do you mean "extend the simulation time by adding the values of fields at the last time step of FDTD simulation"?

**ANSWER:**
Thank you for taking your time to review our manuscript.
We are sorry for being not clear in this description, and we would like to clarify this part of our analysis. Using the computer cluster we were able to run FDTD simulations up to 100 s, which gives the frequency resolution $df = 0.01$ Hz. Next we were trying to improve the frequency resolution by extending the simulation manually to 1000 s by adding the values of EM fields at the last time step of FDTD simulation (these values at $t = 100$ s are almost zero). And that what we meant by "extend the simulation time".
However, we realized that such operation is not really improving the frequency resolution in case of strong attenuation of electromagnetic waves, which is our case.
After discussing this issue in our team and following the reviewers' suggestions we have decided to remove this manual extension of TD result and keep the original (*true*) $t_{max} = 100$ sec, and $df = 0.01$ Hz.

**REVIEWER COMMENT:**

> Line 120 – 130, please add more details for the filter used in the lightning source and add the comparison results between the original source and the filtered one.

**ANSWER:**
Thank you for this comment. We have added the details of filtering that we have applied. Also in Figure 1 below we present TD and FD results for filtered and not-filtered source. On that plot we show the result for probe position at $\theta = 10°$.
Filtered and not-filtered results are similar in FD in Schumann resonance range: almost identical for the magnetic field $H_p$, but in case of electric field $E_r$ for not-filtered source there are some oscillations in the spectra. Not-filtered source introduces strong artefacts in the time domain and also and in the frequency domain at frequencies out of the range of interest. Lack of filtration would also lead to energy leakage, which would influences the amplitude.

**REVIEWER COMMENT:**

> Line 131, instead of mention "a finite speed", please give the value of the velocity used in the calculation.

[Figure]

Figure 1: The comparison of TD and FD results for filtered and not-filtered source.

**ANSWER:**
Thank you for this comment. We have added the value of the velocity, that we took from Rakov (2007), namely $v = 10^8\, m/s$.

**REVIEWER COMMENT:**

> The Schumann resonance spectrum figures corresponding to the table 1-3 are suggested to present in the paper.

**ANSWER:**
Thank you for this suggestions. We have added the correspondent spectra to all models of conductivity profiles that we have used.

**REVIEWER COMMENT:**

> In section 3.3, I don't really understand why the authors used the "two-layered" waveguide since there are different exponential profiles which are capable to describe the features. For example, the traditional Wait and Spies' exponential profile gives a reasonable approximation for the electron density profile below 90-km altitude compared to the IRI model. How did you deal with the conductivity located at the boundary between the different two layers?

**ANSWER:**
We have used two-layered conductivity profile in order to be able to validate the FDTD simulations, because for two-layered profile we can calculate the results analytically using our analytical model, and then compare it with FDTD.
As a continuation of our FDTD simulations using more realistic conductivity profile we decided to use recently published profile from Kudintseva et al. (2016). Since this profile is defined up to 100 km, the electromagnetic waves were not attenuated enough, causing artefacts. Therefore we have decided to extend the conductivity profile from Kudintseva et al. (2016) with some realistic continuation to higher values of conductivity, and as such natural continuation we have used the corresponding IRI profile for middle altitudes which smoothly extends the profile from Kudintseva et al. (2016).

**REVIEWER COMMENT:**

In section 4, I am also confused, instead of the entire IRI model, why did the authors use the partial profile proposed in Kudintseva et al, 2016] and partial IRI model? Please add more comments for this part.

**ANSWER:**
We needed a realistic model from the Earth's surface up to about 500 km. Since IRI model is defined for $h > 80$ km we combined it with a recently published conductivity profile in Kudintseva et al. (2016) for the altitudes $h < 100$ km.

**Minor comments:**

**REVIEWER COMMENT:**

Line 25, add the reference:
Marshall, R. A. (2012), An improved model of the lightning electromagnetic field interaction with the D-region ionosphere, J. Geophys. Res., 117, A03316, doi:10.1029/2011JA017408

**ANSWER:**
We have added this reference.

**REVIEWER COMMENT:**

Line 42, the full-wave method (FWM) based on finite element approach need to mention in the introduction part:
Lehtinen, N. G., and U. S. Inan (2008), Radiation of ELF/VLF waves by harmonically varying currents into a stratified ionosphere with application to radiation by a modulated electrojet, J. Geophys. Res., 113, A06301, doi:10.1029/2007JA012911
Lehtinen, N. G., and U. S. Inan (2009), Full-wave modeling of transionospheric propagation of VLF waves, Geophys. Res. Lett., 36, L03104, doi:10.1029/2008GL036535

**ANSWER:**
We have added these studies to the introduction.

**REVIEWER COMMENT:**

Line 44, if I understand here correctly, the authors want to mention the "Q bursts" in the Schumann resonance:
Ogawa, T., Y. Tanaka, M. Yasuhara, A. C. Fraser-Smith, and R. Gendrin. "Worldwide simultaneity of occurrence of a Q-type ELF burst in the Schumann resonance frequency range." Journal of geomagnetism and geoelectricity 19, no. 4 (1967): 377-384.

**ANSWER:**
We have added this reference.

**REVIEWER COMMENT:**

Line 142, please explain the parameters in the equation there.

**ANSWER:**
Thank you for this comment. We have added the explanation of these parameters.

---

## Referee Report (RR1)

After looking through the text and through the replies to both reviews, I admit that considerable work has been done. Now, the paper could be sent for publication in its present form.

Still, I cannot help myself from a few remarks.

I had much fun when reading the materials. The major problem of authors is a poor knowledge of SR literature in spite they mention that were involved in the field for the long time. This is why they state that Greifingers (1978) used the exponential model while Greifingers applied the Pierce and Cole profile in their computations/. They introduced the both characteristic heights by using the localized approximating exponential sections in the vicinity of these two areas of high power losses in the ionosphere.

Authors apply the two layer model and do not mention that such a model was suggested by David Jones in 1964 who had found its parameters from the SR observations.

There are some odd results passed unnoticed by authors. I will mention only two of them.

Manuscript states that the conductivity profile must be postulated at heights up to a few hundreds of kilometers. Simultaneously, the left frame in Fig. 4 clearly indicates that the upper characteristic height never reaches the 100 km altitude. However, this contradiction does not bother the authors.

In the revised manuscript, the TD solutions are shown. The outline of pulses computed for the considerable source – observer distances looks rather good as well as corresponding spectra, provided that we do not go into details and compare these with observations. The real problem is hidden in the left frames of Fig. 1 where the ideal cavity is treated. I would drive attention of authors to the paper [Nickolaenko, A. P., and M. Hayakawa (2014), Spectra and waveforms of ELF transients in the Earth-ionosphere cavity with small losses, Radio Sci., 49, doi:10.1002/2013RS005281] and indicate that outline of their spectra has nothing in common with that shown in this work. It is interesting in this connection, in what a way the spectra were obtained at all? The amplitudes are infinite at the resonance frequencies by definition, and I cannot even imagine how one could initially compute the waveforms in a perfect cavity. Certainly, Figure 1 in the Response to the second Reviewer is incorrect. Look at the plot in the middle of it. The first pulse is the direct wave from a positive stroke. The second pulse must be the antipodal wave, however, what had happened to the waveform? The third pulse is the first round-the-world wave, but why the initially positive stroke had turned into the negative one?

A separate question remains, in what a way the high frequency resolution was obtained in the work while the temporal realizations do not exceed 0.2 s? One must compute the TD record of 10 s duration to have the 0.1 Hz resolution. To what an extent the pulse amplitude will be reduced in this case?

Nevertheless, I will not repeat my comments of the first review. Some of these were taken into account. The above mentioned facts tell a specialist that there are serious unresolved problems in the approach applied in the work.

I think that authors have done the best they could, so that paper might be published in its present form.

Yours sincerely,
A.P. Nickolaenko